# Extreme heat prediction through deep learning and explainable AI

**Fatima Shafiq**[1�], **Amna Zafar**[1�], **Muhammad Usman Ghani Khan**[1], **Sajid Iqbal**[iD][2]*, **Abdulmohsen Saud Albesher**[2‡], **Muhammad Nabeel Asghar**[iD][2‡]

**1** Department of Computer Science, University of Engineering and Technology, Lahore, Punjab, Pakistan, **2** Department of Information Systems, College of Computer Science and Information Technology, King Faisal University, Saudi Arabia

‡ FS collected data, performed data prepossessing and conducted experiments, AZ conceived the research idea, performed literature review, designed the experimental setup, UG, SI, ASA, and MNA performed write up and fine tuned the experiments, cross check the codes.
☍ These authors contributed equally to this work.

* siqbal@kfu.edu.sa

**Data availability statement:** The data underlying the results presented in the study are available from Ms. Fatima Shafeeq at (https://github.com/FatimaaShafiq/XWeather-Dataset/tree/main.

**Funding:** This work was supported by the Deanship of Scientific Research, Vice Presidency for Graduate Studies and Scientific Research, King Faisal University, Saudi Arabia, under Project GrantA468.

**Competing interests:** No authors have competing interests.

## Abstract

Extreme heat waves are causing widespread concern for comprehensive studies on their ecological and societal implications. With the ongoing rise in global temperatures, precise forecasting of heatwaves becomes increasingly crucial for proactive planning and ensuring safety. This study investigates the efficacy of deep learning (DL) models, including Artificial Neural Network (ANN), Conolutional Neural Network (CNN) and Long-Short Term Memory (LSTM), using five years of meteorological data from Pakistan Meteorological Department (PMD), by integrating Explainable AI (XAI) techniques to enhance the interpretability of models. Although Weather forecasting has advanced in predicting sunshine, rain, clouds, and general weather patterns, the study of extreme heat, particularly using advanced computer models, remains largely unexplored, overlooking this gap risks significant disruptions in daily life. Our study addresses this gap by collecting five years of weather dataset and developing a comprehensive framework integrating DL and XAI models for extreme heat prediction. Key variables such as temperature, pressure, humidity, wind, and precipitation are examined. Our findings demonstrate that the LSTM model outperforms others with a lead time of 1–3 days and minimal error metrics, achieving an accuracy of 96.2%. Through the utilization of SHAP and LIME XAI methods, we elucidate the significance of humidity and maximum temperature in accurately predicting extreme heat events. Overall, this study emphasizes how important it is to investigate intricate DL models that integrate XAI for the prediction of extreme heat. Making these models understood allows us to identify important parameters, improving heatwave forecasting accuracy and guiding risk-reduction strategies.

## 1 Introduction

Extreme heat refers to deviations from normative high temperatures within a specific geographic location, determined by unique meteorological conditions. The World

Meteorological Organization classifies meteorological heatwaves into dry and wet types, with dry heatwaves characterized by clear skies and oppressive temperatures and wet heatwaves characterized by humid temperatures [1]. As one of the main contributors to both natural and human disasters, heatwaves must be addressed, and preventive measures must be taken to lessen the damage [2]. Extended heatwaves can cause far more havoc than a single, severe day's temperature [3]. Global cities are the primary sources of heat and pollution due to factors like energy use and economic activities [4]. In 2010, there was an atmospheric blockage in Eastern Europe and Western Russia [5] that resulted in over a month of intense heat and 1500 tragic deaths. High temperatures pose significant threats to civilizations due to their potential impact on various economic aspects of living [6]. A heatwave that struck Karachi in June 2015 caused catastrophic damage and 1200 deaths. The combination of low pressure and high humidity resulted in a high temperature of 44.8 °C on June 20, 2015 [7]. Serious concerns regarding the 21st century arise from global changes in weather patterns [8]. In this paradigm, Pakistan find itself more a victim than a catalyst of these climate changes [9]. The summer months in Pakistan typically run from March to August, when temperatures range from mild to hot. However, during the past three decades, Pakistan has seen an increase in severe weather events [10] comparable to the vulnerabilities found in coastal Ecosystems especially vulnerable to oil spills and human activities, like mangroves and leatherback turtle nesting sites along Trinidad's east coast [11], highlight the need for strategic planning to minimize environmental damage and preserve ecological integrity and public health. The health of its people, agriculture, and the general ecological balance are all seriously threatened by this expansion. Workers in construction companies in Al-Ahsa region of Saudi Arabia are at a higher risk of kidney injury due to extreme summer heat, which is worsened by dehydration, a lack of sleep, and obesity. This highlights the broader effects of heatwaves across different regions [12].

Scholars are utilizing technological advancements like deep learning (DL) to address the ever-changing climate patterns, especially during periods of extreme heat. LSTM, a DL models have garnered attention for weather forecasting [13–16]. However, even with these developments, interpretability is still a critical aspect of DL models. Understanding and interpreting the algorithm's outcomes is crucial [17]. In this case, XAI is useful because it improves transparency by offering insights into the results of the DL model and revealing the factors that underpin the heat prediction. By utilizing XAI's capabilities, stakeholders can gain a comprehensive understanding of the model's predictions, which promotes confidence and understanding in the model's decision-making process. Many deep learning (DL) and machine learning (ML) models have been used to forecast extreme heat events; however, these models are mainly used to forecast weather events with different lead times, such as forecasts for the short and long term [7,18,19,21]. To the best of our knowledge, minimal research has been done to investigate the crucial elements required to make precise predictions [22]. It is imperative that stakeholders comprehend these factors in order to take proactive measures aimed at mitigating the effects of extreme heat events. In this work, we collected regional data from the Pakistan Meteorological Department (PMD) and, after preprocessing, named it the XWeather (eXplainable Weather) dataset, which serves as the foundation for our research. Leveraging this dataset, we propose a framework that uses DL models to predict extreme heat and XAI to provide insights into the contributing factors. In particular, the contributions of this study are:

- Preparation of XWeather dataset by collecting and pre-processing 5 years of meteorological data (2018-2022) sourced from PMD, which includes label encoding and scaling of numerical features

- We proposed a framework utilizing three deep learning models including ANN, CNN, and LSTM to predict extreme heat events, with LSTM achieving the highest accuracy in forecasting these events.
- Applying SHAP and LIME XAI to extract high impact features for precise heat event prediction. By pinpointing and justifying the most critical features, we effectively filter out low-value data, significantly boosting model accuracy and reliability.

Our study focuses on integrating explainable AI principles with deep learning methodologies to provide transparent insights into factors driving extreme heat event predictions, to enhance decision-making and mitigation techniques. The remaining structure of the paper is as follows: In Sect 2, a thorough related work on DL, ML, and statistical models for extreme heat is reviewed. In Sect 3, we present a comprehensive framework that includes details of the XWeather dataset preprocessing and the functioning of both DL and XAI models. Sect 4 describes the experimental setup, covering hardware and software specifications as well as the training procedures for the DL models. Sect 5 provides an in-depth discussion of the results, highlighting key insights and analyzing the performance of the proposed models. Finally, Sect 6 concludes the study by summarizing the findings and suggesting possible directions for future research.

## 2 Literature review

Extreme heat prediction is critical for its serious effects, necessitating extensive global research using a variety of methodologies such as ML, DL, and statistical models. But interpreting and understanding the variables influencing these forecasts is a challenging task. Integrating XAI techniques has the potential to elucidate critical features in models, improve interpretability, and formulate proactive measures to mitigate heat wave effects. This section covers the latest research on the forecasting of extreme heatwaves in different parts of the world. In study [22] Sub-seasonal performance analyzed in Central Europe using linear and RF models, revealing accurate ECMWF forecasts with 1-6 weeks lead times. Local air temperature, 500 hPa soil temperature, and precipitation, were identified as the greatest predictors of summer anomalies and heat waves. The temperature data from 27 grid points are used in another study [23] to forecast the number of annual heatwave days in Iran. Ada-Boost Regression and Decision Tree (ABR-DT), a hybrid approach that combines traditional machine learning algorithms and hybrid methods, performs better than the other methods with a correlation coefficient of 0.860 and a mean absolute error of 6.929. The research [19] develops a climate change-resilient heatwave prediction model using machine learning methods like SVM, random forest, and artificial neural network, outperforming other models in Pakistan summer heatwave days. A climate change study on Trinidad and Tobago using Sen's slope estimator and Mann-Kendall tests found significant increases in minimum and maximum temperatures over 30 years, with no significant changes in monthly rainfall. This suggests warmer winters and hotter summers [20].

The neural weather model, ExtNet [25], outperforms historical data in predicting heat events with lead time of 1 to 28 days in sub seasonal conditions. The next study [26] assesses the prediction of summer high temperatures in Western and Central Europe using statistical modeling and a machine learning model on the ERA5 dataset, with a 15-day lead time, concentrating on climate change and driving variables. The study [24] compares ANN and Gene Expression Programming (GEP) for temperature prediction in Tabuk, Saudi Arabia. GEP, using inputs like pressure, wind speed, humidity, and rainfall, proves more accurate

and reliable for agricultural forecasting. However, in study [21] Convolutional Neural Network trained using 1000 years of climate model results, Planet Simulator, large-class under-sampling, and transfer learning, accurately anticipates long-lasting intense heat waves up to 15 days in advance. A real-time heatwave monitoring method [27] for the Indian sub-continent, utilizing machine learning to predict extreme heat indices two to three days in advance. The SWAT (Soil and Water Assessment Tool) model in [29] showed that climate change could increase extreme heat temperatures by 1–4 °C in the Gharesou Basin, affecting water availability. The QRF model, based on synoptic climate variables, outperforms classical random forest in Pakistan's heatwave forecasting, predicting triggering and departure dates with 1–10 days lead times [30]. The LSTM neural networks [7] is utilized to predict heatwave maximum temperatures. Moreover, a data-driven solution [31] using Capsule Neural Networks (Caps Nets) for weather forecasting extremes, outperforming numerical models and demonstrating the power of deep learning methods. However, the study [32] explores climate change's impact on the China-Pakistan Economic Corridor, revealing its vulnerability to heat waves from 1980 to 2016, with continuous activity in Pakistan's southern, middle, and eastern regions. The study [18] employs Recurrent Neural Networks and LSTM models to estimate daily temperatures from BMKG in Bandung, with accuracy enhanced by pre-processing and optimization models. The study [28] forecasts temperature and geopotential approximately three to five days in advance using a full stacked neural network approach, specifically convolutional ResNet over the ERA5 dataset. The study [1] reveals a global concentration of heatwave research and health impacts, primarily in mid-latitude, high-income countries, with underrepresentation in areas like tropical and high-latitude areas, South America, Africa, Eastern Europe, Middle East, and Asia. It is anticipated that in the near future, South Asian regions and its neighbors would be extremely affected by HWs, which will have significant socioeconomic effects [32].

**Limited work exists on regional datasets:** In the field of extreme heat prediction, there has been relatively little focus on datasets specific to south Asian regions [7,19,27] despite this being a critical and emerging area of interest. The complexity of acquiring localized data often poses a significant barrier, which may explain the scarcity of such studies. To address this gap, we collected regional data from the PMD, preprocessed it, and named it the XWeather dataset. To the best of our knowledge, limited work has utilized local datasets in this context, highlighting the novelty and potential impact of our approach.

**Limited work on explainable AI:** Even though XAI is increasingly being used to improve model interpretability, its applications in extreme heat prediction are limited [21–23,25,33, 34] . Given the importance of making accurate and interpretable predictions in this field, there is an increasing need for XAI to help unpack the "black box" nature of deep learning models. To close this gap, we use XAI methods to gain insights into how predictions are made and to identify the critical factors driving extreme heat events, making a significant contribution to model transparency and reliability in this field.

**Limited work on deep learning with XAI:** DL models are complex [35], [36] and frequently known as "black boxes" making it difficult to comprehend their prediction processes. While much work has been done to integrate XAI with machine learning models [26,37], there has been little research into applying XAI to DL models, particularly in the context of extreme heat prediction. This study fills that gap by utilizing XAI to identify the factors influencing DL model predictions, thereby improving interpretability and reliability in this domain.

To the best of our knowledge, no prior research has looked into the extraction of important features for extreme heat prediction using XAI as shown in Table 1, particularly in combination with DL models. In order to close this gap, we have created a framework in this paper that focused on using DL models including ANN, CNN and LSTM for extreme heat prediction, with an emphasis on the Asia region. In order to improve the transparency and interpretability of DL-based extreme heat prediction models, our method places a strong emphasis on the extraction of important features using XAI techniques.

## 3 Proposed methodology

The proposed methodology for heat prediction includes the following steps:

- Collection of XWeather dataset containing relevant meteorological parameters from PMD.
- Preprocess the data by handling missing values, per forming numerical feature scaling, and applying label encoding.
- DL Model training including ANN, CNN, and LSTM.
- Evaluate the models using metrics such as accuracy, RMSE, MAE, and MSE.
- Select the model with higher accuracy and minimal error rates for establishing regional thresholds.

**Table 1. Comparative analysis of different approaches used for extreme heat prediction.**

| Ref | Dataset Collection | Area of Study | Technique | Lead Time Findings | XAI |
|---|---|---|---|---|---|
| Weirich Benet et al. (2023) | ECMWF | Central Europe | Linear and Random Forest | 1-6 weeks | × |
| Lopez Gomez et al. (2023) | ERA5 | Globally | Neural Weather Models | 1 to 28 days | × |
| Van Straaten et al. (2022) | ERA5 | Central and Western Europe | Logistic Regression | 15 days | ✓ |
| Juna et al. (2022) | World Weather Online | Karachi, Pakistan | LSTM Deep Learning | × | × |
| Dumas et al. (2022) | PlaSim climate model | France | CNN Deep Learning | 15 days ahead | × |
| Narkhede et al. (2022) | Indian Meteorological Department (IMD) | India, Asia | Empirical Model/Vanilla LSTM model | 2–3 days | × |
| Anushka Perera et al. (2022) | Trinidad and Tobago Meteorological Service | Trinidad, Tobago Islands | Sen's slope estimator and Mann-Kendall tests. | × | × |
| Khan et al. (2021) | NCAR/NCEP | Pakistan, Asia | ANN, SVM, Random Forest | × | × |
| C. Clare et al. (2021) | ERA5 | Globally | Convolutional ResNet, Neural Network | 3–5 days | × |
| S. Rahayu et al. (2020) | BMKG Agency | Indonesia | RNN, LSTM Deep Learning | × | × |
| Chattopadhyay et al. (2020) | large-ensemble (LENS) Community Project | North America | CapsNets Deep Learning | 1–5 days | × |
| Khan et al. (2019) | Princeton Global Forcing/NCEP | Pakistan, Asia | Quantile Regression Forest | ±5 days | × |
| Ullah et al. (2019) | PMD | China-Pakistan Economic Corridor (CPEC) | Mann-Kendall (m-MK) test and Theil-Sen's (TS) test | × | × |
| H. Md. Azamathulla et al. (2018) | Saudi General Authority of Meteorology and Environmental Protection | Tabuk, Saudi Arabia | ANN and GEP | × | × |

**Notes**: XAI = Explainable Artificial Intelligence; ECMWF = European Centre for Medium-Range Weather Forecasts; ERA5 = Fifth generation of ECMWF atmospheric reanalyses of the global climate; LSTM = Long Short-Term Memory; ANN = Artificial Neural Network; SVM = Support Vector Machine; RNN = Recurrent Neural Network; CNN = Convolutional Neural Network.

- Incorporate explainable AI models including SHAP and LIME to ensure prediction transparency.

These steps (see Fig 1) contribute to a comprehensive approach for predicting heat events and identifying the critical elements that should guide mitigation and management plans.

## 3.1 Dataset details

This study focuses on Lahore, the capital of Pakistan's Punjab province, which covers about 1772 square kilometers. It experiences hot summers with highs up to 40 °C (see Fig 2), and chilly winters with moderate to heavy rainfall. Extreme weather events have a significant impact on the city's infrastructure and inhabitants due to industrial activities and widespread vehicle use exacerbating environmental problems like air and water pollution. For this reason, the XWeather dataset, collected from PMD, spans over five years from 2018 to 2022 and comprises twelve parameters, including date, pressure, humidity, minimum and maximum temperatures, precipitation, wind speed, and wind direction, recorded at 300 and 1200 UTC (Universal Time Coordinated). See Table 2 for details.

**3.1.1 Preprocessing** The XWeather dataset used in our study was carefully preprocessed to reduce biases and extract insightful information that would help us make more accurate predictions for extreme heat events. Label encoding, numerical feature scaling, and missing data configuration were applied as preprocessing techniques. It includes;

- **Handling missing data:** The average values of the corresponding column were utilized to ensure the accuracy and completeness of the subsequent analyses

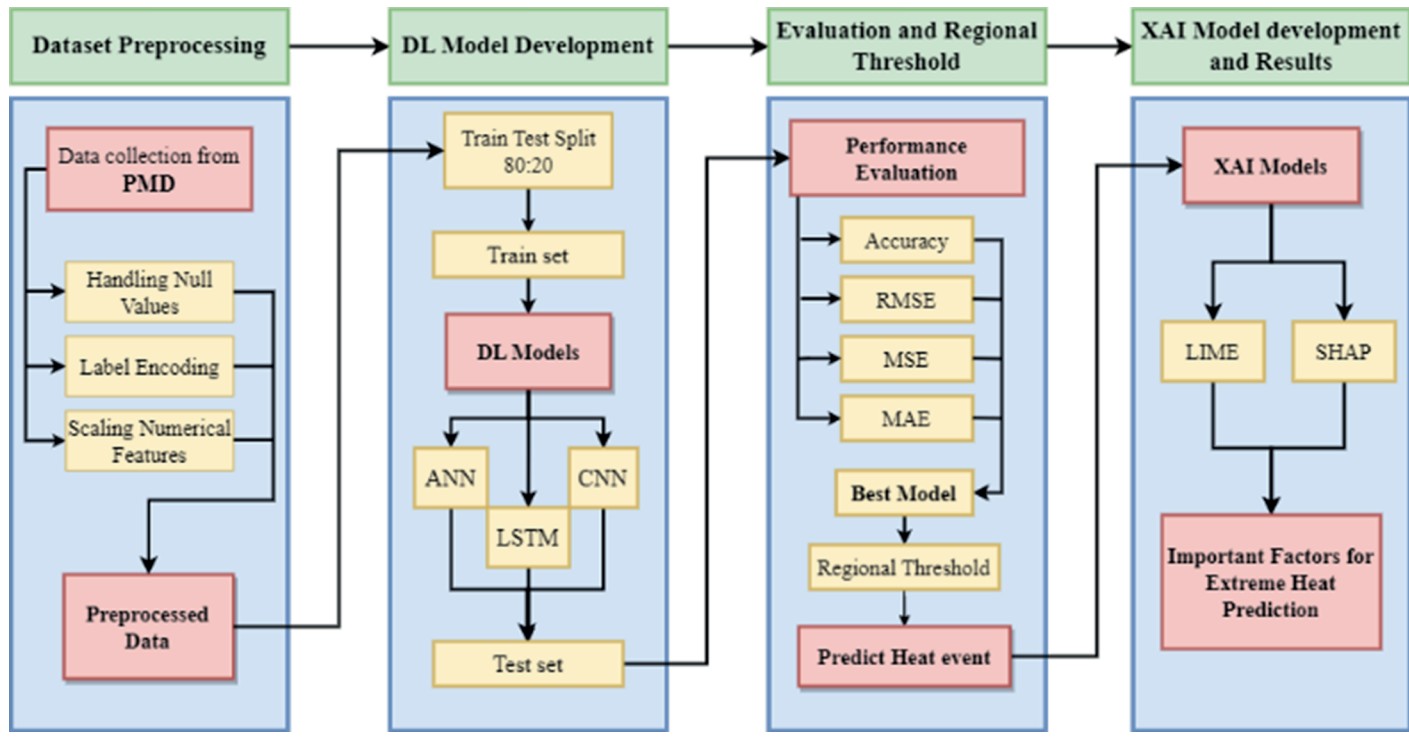

**Fig 1. Proposed methodology for predicting extreme heat events.**

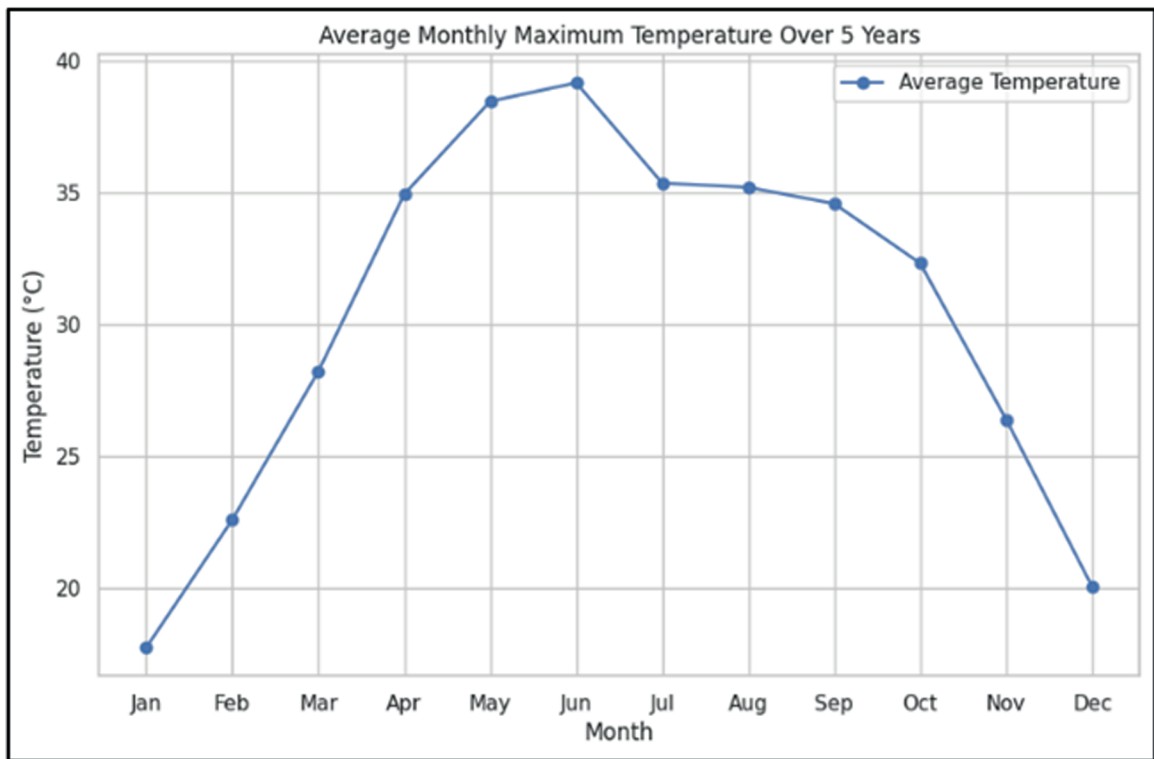

**Fig 2. Monthly average maximum temperature of study region over the past 5 years.**

**Table 2. Details of XDL dataset parameters.**

| Sr. # | Parameters | Daily resampling | Units |
|---|---|---|---|
| 1 | Date | Daily | d-m-y |
| 2 | Pressure | 300 and 1200 UTC | Millibars (mb) |
| 3 | Humidity | 300 and 1200 UTC | g/kg |
| 4 | Minimum Temperature | 24h | Celsius |
| 5 | Maximum Temperature | 24h | Celsius |
| 6 | Wind Direction | 300 and 1200 UTC | Cardinal |
| 7 | Wind Speed | 300 and 1200 UTC | Meter per second |
| 8 | Precipitation | 24-h mean | millimeter |

- **Label encoding:** The process of converting categorical variables like wind direction into numerical representations by assigning distinct integer labels for each wind direction observed at 300 and 1200 UTC.
- **Numerical feature scaling:** Our data underwent min max scaling, a process of subtracting the minimum value and dividing it by the data range, resulting in values typically falling within a predefined range.

This ensured that complex weather patterns that cause periods of intense heat were appropriately represented in XWeather dataset. A correlation heatmap (see Fig 3) provides a visual representation of the relationships between variables The degree of correlation between variables is indicated by the color intensity in a correlation heatmap, as demonstrated by the matrix with color coding shown below.

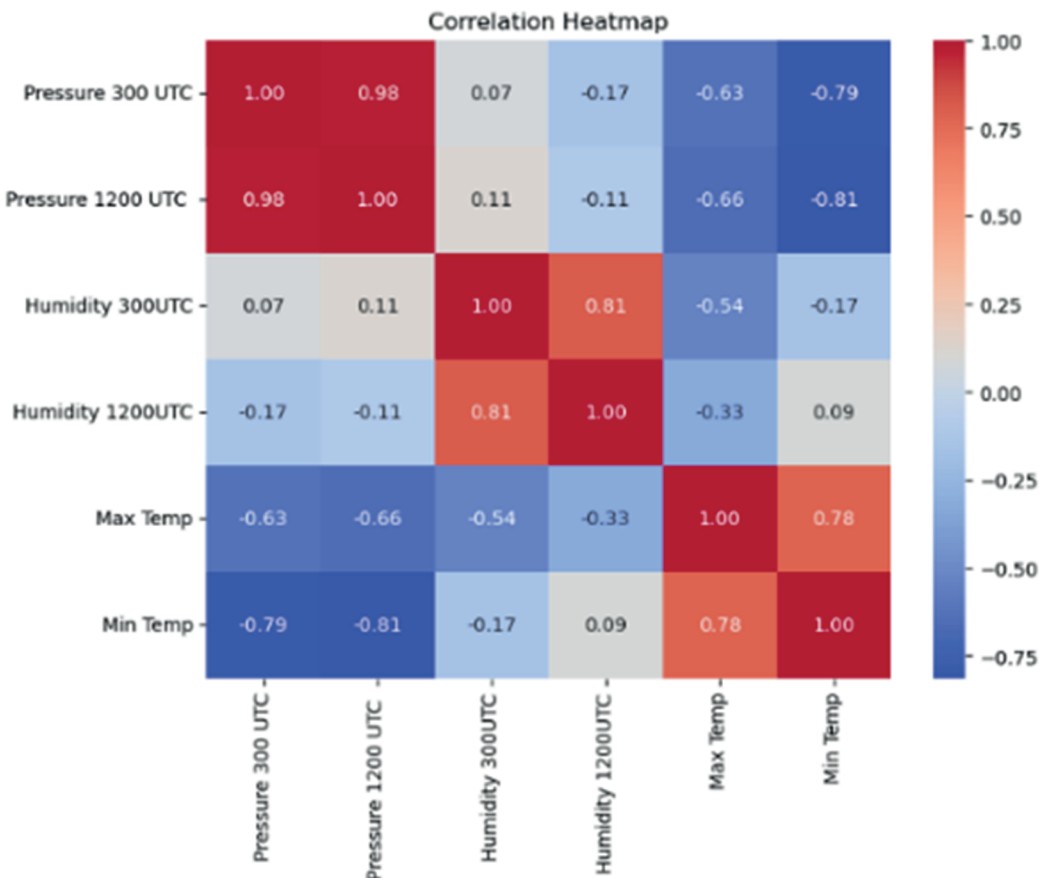

**Fig 3. Correlation Heatmap of XWeather dataset parameters.**

## 3.2 Deep learning model development

Deep learning is an integral branch of artificial intelligence that has become very popular due to its ability to solve complicated time series problems. In this study, extreme heat events are subjected to predictive analysis using deep learning models including ANN, CNN, and LSTM that are capable of automatically learning from data. Utilizing the deep learning packages, including Keras and TensorFlow, all code is written in the Python programming language

**3.2.1 ANN model** In this study, the architecture of our model (see Fig 4) consists of three hidden layers, each with 64 neurons acti vated by a rectified linear unit (ReLu) function. The second layer maintains 32 neurons, while the third layer integrates a single neuron acti-vated by a sigmoid function. To optimize the model, we used a batch size of 32 during train-ing over 50 epochs, balancing computational efficiency and model convergence. This struc-tured approach ensures clarity and precision in our neural network architecture and training regimen.

**3.2.2 CNN model** In our study, CNNs serve as the secondary algorithm, showing an exemplary architecture tailored for handling intricate datasets. Illustrated in Fig 5 our model features a convolutional layer comprising 16 filters, each paired with a ReLu activation func-tion, succeeded by a pooling layer for effective down-sampling. Subsequently, data is pre-pared for subsequent dense layers through a flatten layer, leading into a 10-neuron dense

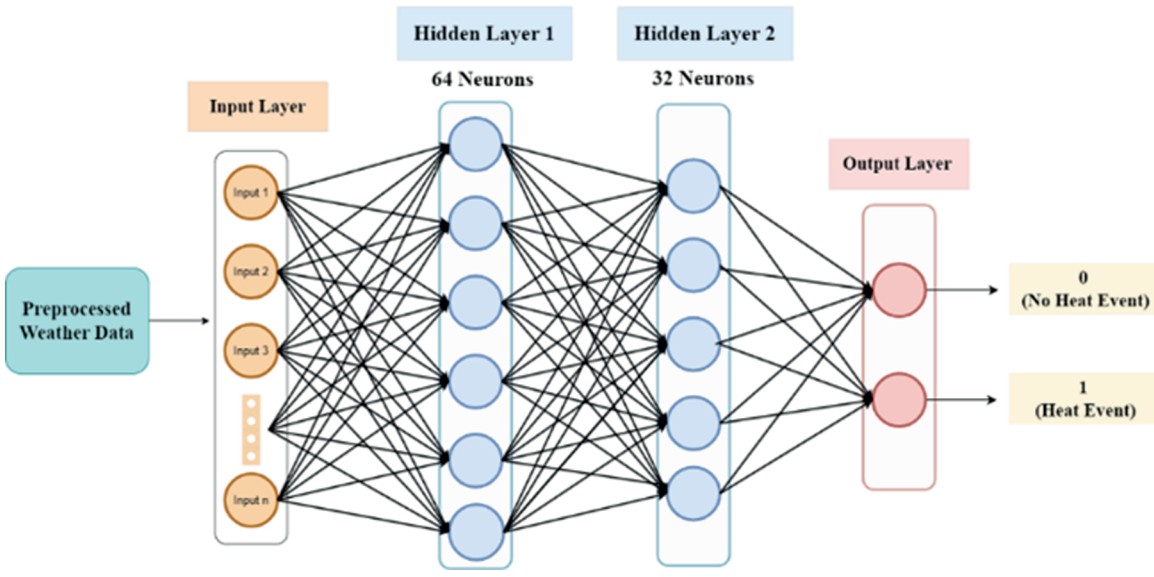

**Fig 4. ANN structure for extreme heat prediction.**

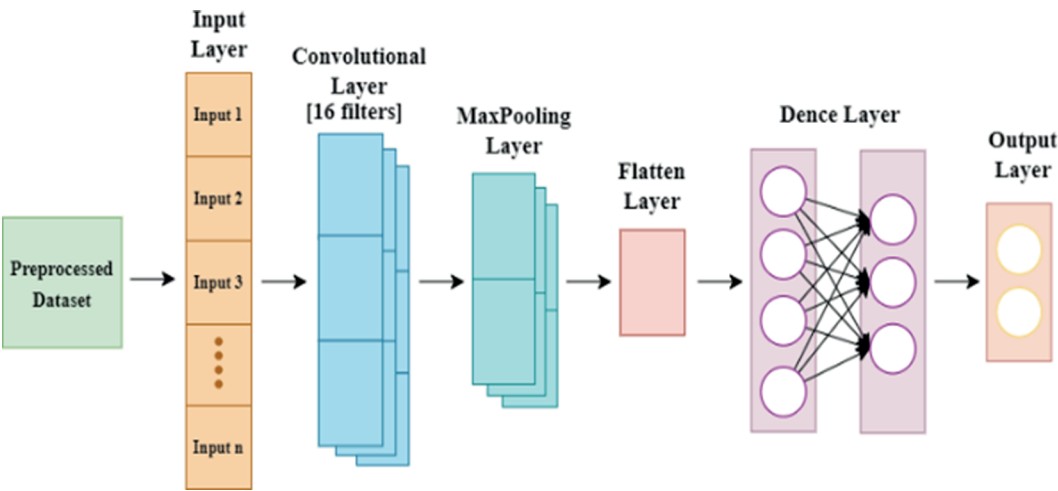

**Fig 5. CNN structure for extreme heat prediction.**

layer equipped with ReLU activation. For binary classification tasks, the final layer harnesses the Sigmoid function. Employing the Adam optimizer, our model is meticulously trained over 50 epochs, utilizing a batch size of 32. The CNN framework emerges as indispensable, adept unveiling intrinsic patterns embedded within datasets, thus enriching our analytical endeavors.

**3.2.3 LSTM model** In essence, LSTM is a well-known, significant advancement of the Recurrent Neural Network RNN that tackles its conventional issues. LSTM is a subtype of RNN with a more complex structure and specialized memory cells with gating mechanisms. The core LSTM structure (see Fig 6) is made up of three gates including input gate, forget gate and output gate. Within a time horizon, each gate regulates and secures the data in both short- and long-term memory. It makes use of both hyperbolic tangent activation function

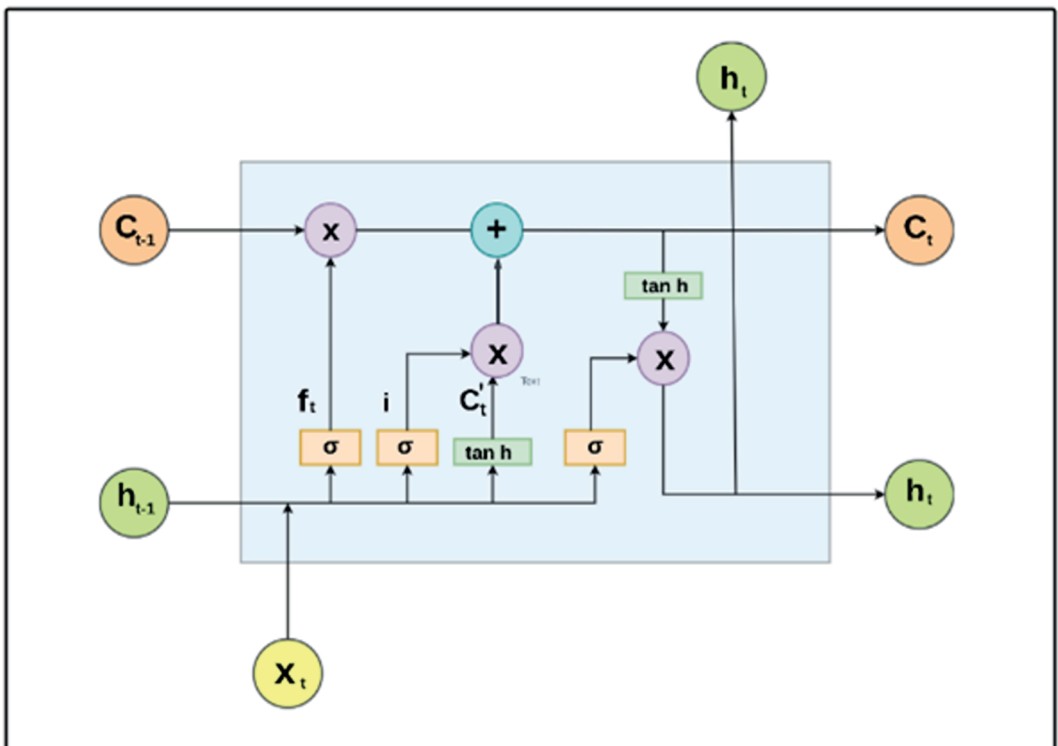

**Fig 6. Detailed gate structure of LSTM architecture reveals intricate insights into long short-term memory networks.**

and sigmoid activation function. Any x-axis coordinate between 0 and 1 is handled by the sigmoid activation function. On the other hand, any x-axis coordinate can be converted to a y-axis coordinate using the hyperbolic tangent activation function.

$$f_t = \sigma \left( W_{hf} h_{t-1} + W_{xf} X_t + b_f \right) \tag{1}$$

In Eq 1, the sigmoid function used by the forget gate is represented by $\sigma$, and $f_t$ stands for the forget gate results. The weight matrices and bias term of the forget gate are represented by $W_{xf}$, $W_{hf}$, and $b_f$, respectively.

$$i_t = \sigma \left( W_{hi} h_{t-1} + W_{xi} X_t + b_i \right) \tag{2}$$

$$C_t' = \tanh \left( W_{hc} h_{t-1} + W_{xc} X_t + b_c \right) \tag{3}$$

Similarly, Eqs 2 and 3 indicate that the cell state has been updated with new data as a result of the current input. Thus, $i_t$ and $C_t'$ represent the results of the input gate and the candidate state, respectively. The terms $W_{hi}$, $W_{xi}$, $W_{hc}$, and $W_{xc}$ represent the weight matrices of the input layer, while $b_i$ and $b_c$ represent the bias values of the input gate and the candidate state, respectively.

$$C_t = f_i C_{t-1} + i_t C_t' \tag{4}$$

Eq 4 can be used to determine the long-term memory that is currently updated. The final stage of this LSTM can be updated using the revised long-term memory obtained through

the input gate. Here, $C_t$ and $C_{t-1}$ stand for the cell state value's current and previous states, respectively.

$$o_t = \sigma \left( W_{ho} h_{t-1} + W_{xo} X_t + b_o \right) \tag{5}$$

$$h_t = o_i \cdot \tanh \left( C_t \right) \tag{6}$$

The gate containing the output can be computed using Eq 5, and the most recent update to the short-term memory is attainable using Eq 6. Here, $W_{ho}$, $W_{xo}$, and $b_o$ represent the weights and bias in the output layer, and $X_t$ is the input variable in the LSTM unit. Additionally, the LSTM can read, reset, and update both short- and long-term data using these three gates, which improves the results for time series problems. Our study underscores the pivotal role of LSTM model development due to its remarkable capability in handling time series datasets, a characteristic particularly pertinent to our XWeather dataset.

Our LSTM architecture (see Fig 7) comprises three layers, with the initial layer containing 150 units and employing a ReLU activation function. The subsequent layer features 100 units, utilizing the same ReLU activation function, culminating in a single unit tailored for binary classification tasks. Leveraging a batch size of 32, the Adam optimizer, and an extensive training regimen of 50 epochs, our model aptly demonstrates its prowess in capturing sequential memory patterns. This capability renders it adept at discerning and sequencing temporal dependencies inherent in sequential data, thereby enhancing its utility in various analytical contexts.

## 3.3 Explainable AI

These days, deep learning and machine learning predictions for weather forecasting get clarified by XAI, which is crucial and enlightening at the same time. XAI models doesn't affect the

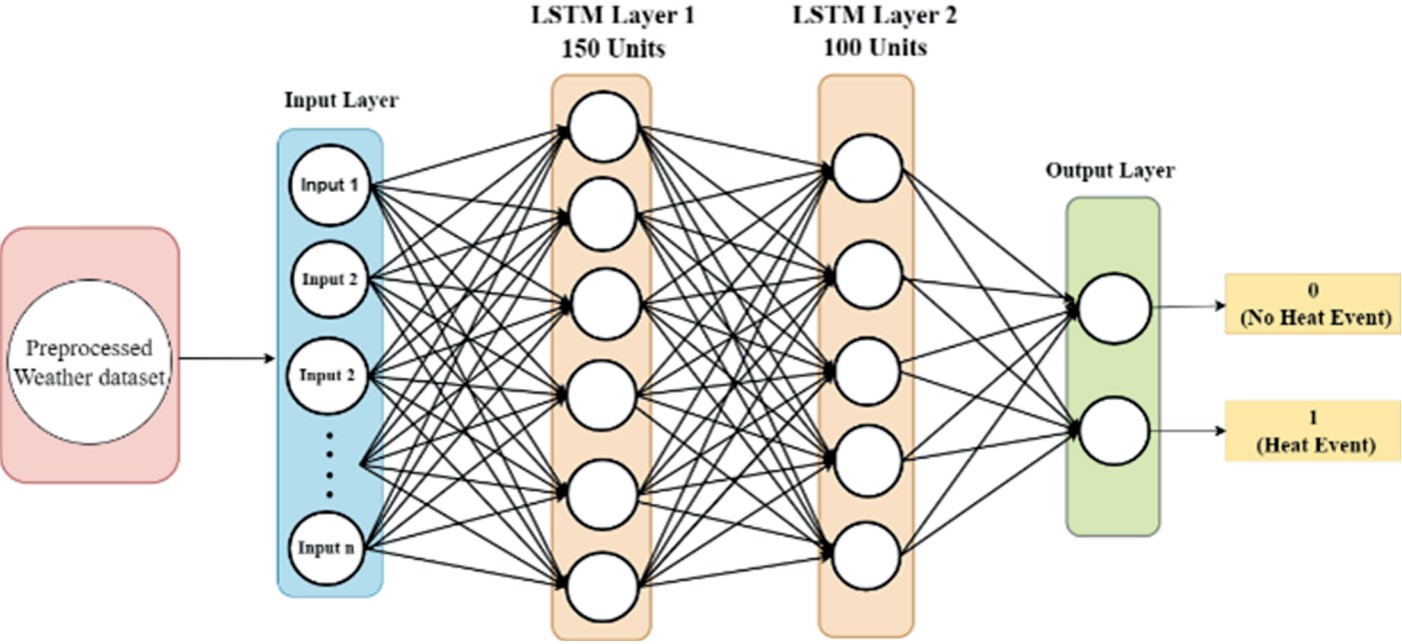

**Fig 7. LSTM structure for extreme heat prediction.**

accuracies and computations of the ML or DL models on which it is used, but it helps in the decision-making process by providing insights into it [38]. While complex black box models [39] might be more accurate than simpler ones, their logic is often unclear. Afterwards explanation techniques are essential in these situations because they offer comprehensible insights into the predictions produced by these complex ML and DL models. These 'black box models' got new transperency level because of XAI techniques like SHAP (Shaply Additive explanations) [40] and LIME (Local Interpretable Model-agnostic explanations). Since DL models are best at forecasting extreme heat events which basically includes time series data, so we apply XAI models to it to gain more insight into these predictions. The SHAP model determines how each parameter contributes to the prediction of heatwaves, whereas LIME determines which factor is most crucial [41].

**3.3.1 SHAP** In this study, the LSTM model results were subjected to the Shapley additive explanation (SHAP) of XAI in order to quantitatively analyze the impact of input variables on the predictions of extreme heat events for every observation period. By examining each input parameter's presence or absence in the dataset, SHAP determines its marginal contribution and assigns a unique importance value to rank each parameter right away. Deep SHAP computes the SHAP values for each component using the compositional nature of neural networks to provide estimates of SHAP insights for the entire network [42,43]. SHAP assigns a distinct value to each observation's parameter by employing the reverse engineering model output method for explaining the reasons behind the predictions [44]. The way the variable combination varies can be used to calculate the Shapley value [45]. A modification to the variable combination occurs when the matching variable is added or removed from the combination. Mathematically,

$$\phi_i(f, x) = \sum_{S \subseteq x} \frac{|S|! \cdot (M - |S| - 1)!}{M!} \left[ f_x(S) - f_x(S \setminus \{i\}) \right] \tag{7}$$

where $f(S)$ is the value generated by the coalition $S$, $M$ is the total number of dataset attributes, $|S|$ is the number of attributes in coalition $S$, and $f_x(S \setminus \{i\})$ represents the value generated when attribute $i$ joins coalition $S$.

This clarifies the degree to which the output value of the learned model is influenced by the input variable. With respect to each input variable, the SHAP values are represented as dots. The vertical axis of the study represents the input parameters, and the horizontal axis represents the impact on the model's output.

$$f(x) = \phi_0 + \sum_{i=1}^{M} \phi_i \tag{8}$$

In the above equation, $\phi_0$ shows the baseline prediction, including the sum of SHAP values for all attributes. The final Shapley value is the sum of the impacts of all the individual predictors, and the SHAP values illustrate how each attribute contributes to it, as shown in Eq 8. The variables on the vertical plane are listed from top to bottom in order of importance based on the range of the SHAP value, which shows the degree of influence on the prediction result. In this case, it was found that humidity had the largest influence on the predicted result, while wind direction had the least. SHAP uses visualization techniques to provide clear understandings of complex deep learning model outputs and actionable insights into their decision-making process.

**3.3.2 LIME** LIME is a technique for defining local explanations for individual complex DL model predictions. This study utilized LIME to quantitatively analyze the impact of input

variables on the LSTM based predictions of extreme heat events, during each observation period. Since LIME is model agnostic, it can be used with any model and is primarily utilized as a local explainer [46,47]. The process involves three steps: initially, the point of interest is surrounded by artificial points in feature space, which are then predicted by the original black-box model, and mapped to a simple model. In order to keep the interpretable model g faithful to the predictions of the complex model f, it undergoes training to reduce a loss function that hinders complexity.

$$L(g) = \sum_{i=1}^{M} w_i \left( f(x_i) - g(x_i) \right)^2 + \Omega(g) \tag{9}$$

In the LIME framework, $w_i$ represents the weight of the altered instances, $f(x_i)$ denotes the prediction made by the complex model (LSTM) for the instance $x_i$, $g(x_i)$ refers to the prediction of the interpretable model for $x_i$, and $\Omega(g)$ stands for the regularization terms used to minimize the complexity of the model $g$. Typically, the interpretable model $g$ is regularized to use a limited number of features, enhancing its comprehensibility. LIME is particularly beneficial for models with a large number of parameters, providing valuable insights into their behavior.

## 4 Experimentation

In our proposed extreme heat prediction model, we initially trained three deep learning models including ANN, CNN, and LSTM on the pre-processed XWeather dataset. The dataset was split into 80% for training and 20% for testing. It contains a total of 1,450 data instances, spanning 5 years. We then applied XAI models, such as SHAP and LIME, to interpret these deep learning models, extracting critical features essential for predicting extreme heat.

### 4.1 Hardware

The experimentations were conducted on an HP laptop having AMD Ryzen 5 PRO 4650U processor with Radeon Graphics running at 2.10 GHz, 8.00 GB of RAM, and a 64-bit operating system (x64-based processor) with Windows 11 Pro. This hardware configuration effectively met the computational requirements for the experiments.

### 4.2 Software

For this research we used open-source libraries and frameworks to implement and train the deep learning models. Python's extensive library ecosystem, including Matplotlib, Pandas, NumPy, and Scikit-learn, facilitated data processing, model training, and evaluation. TensorFlow provided the core functionalities for building and training deep learning models. Additionally, we used SHAP and LIME (including LimeTabularExplainer) to interpret model predictions, extracting key features for a comprehensive understanding of the extreme heat prediction models.

### 4.3 Hyperparameter tuning

To improve the accuracy of each deep learning model for extreme heat prediction, we adjusted model-specific parameters such as the number of hidden layers, learning rate, and regularization settings. Table 3 shows the values chosen for these parameters during training.

### 4.4 ANN model training

In our proposed Extreme Heat Prediction Model, we utilized ANN consisting of three hidden layers. The second hidden layer maintains 32 neurons, while the third layer integrates a single neuron activated by a sigmoid function. This architecture was designed to balance model complexity with computational efficiency. The model was optimized with a batch size of 32 during training over 50 epochs, achieving an accuracy of 89.1%. The training process was completed in 20 epochs, ensuring effective model convergence while balancing training time. Figs 8 and 9 is showing the training and validation loss and accuracy graphs below.

**Table 3. Hyperparameter values for deep learning models.**

| Parameters | Values |
| --- | --- |
| Layers | Dense |
| Activation Function for all Layers | ReLU |
| Activation Function for Output Layer | Sigmoid |
| Total Epochs | 50 |
| Optimizer | Adam |
| Batch Size | 25 |
| Learning Rate | 0.001 |

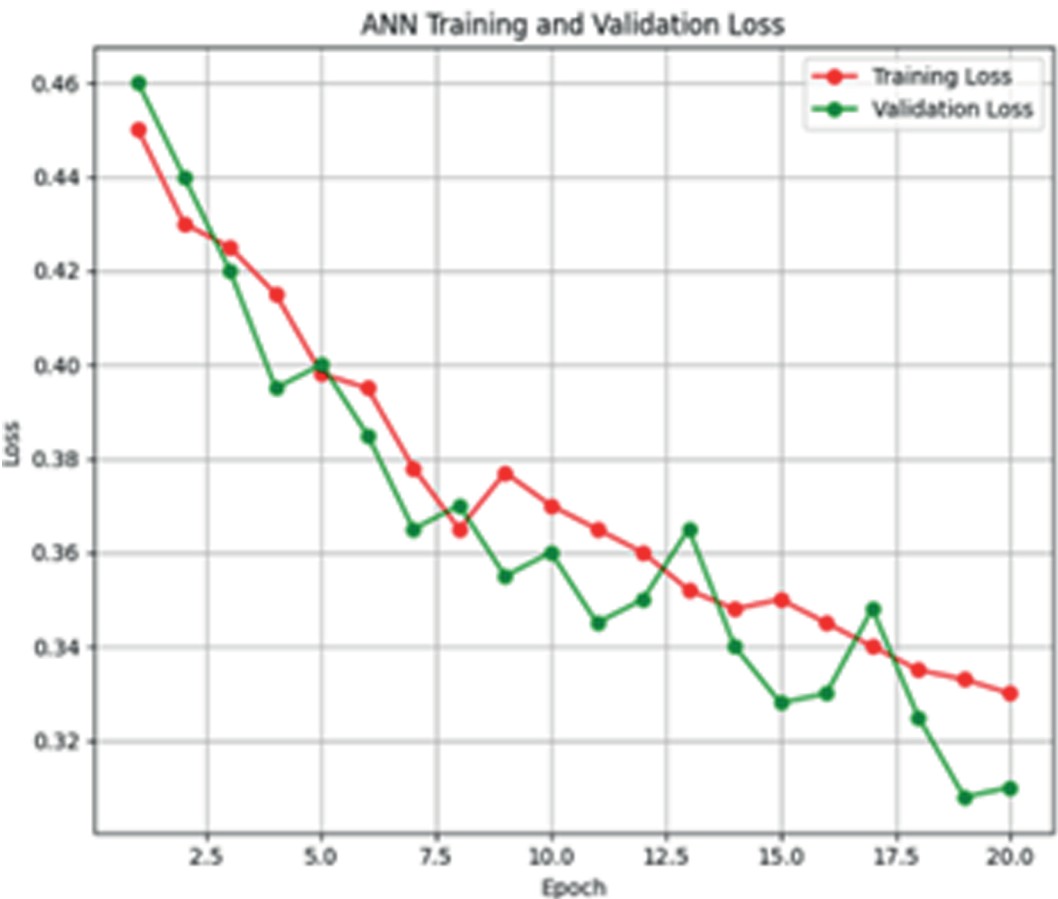

**Fig 8. Training vs. validation loss of ANN model.**

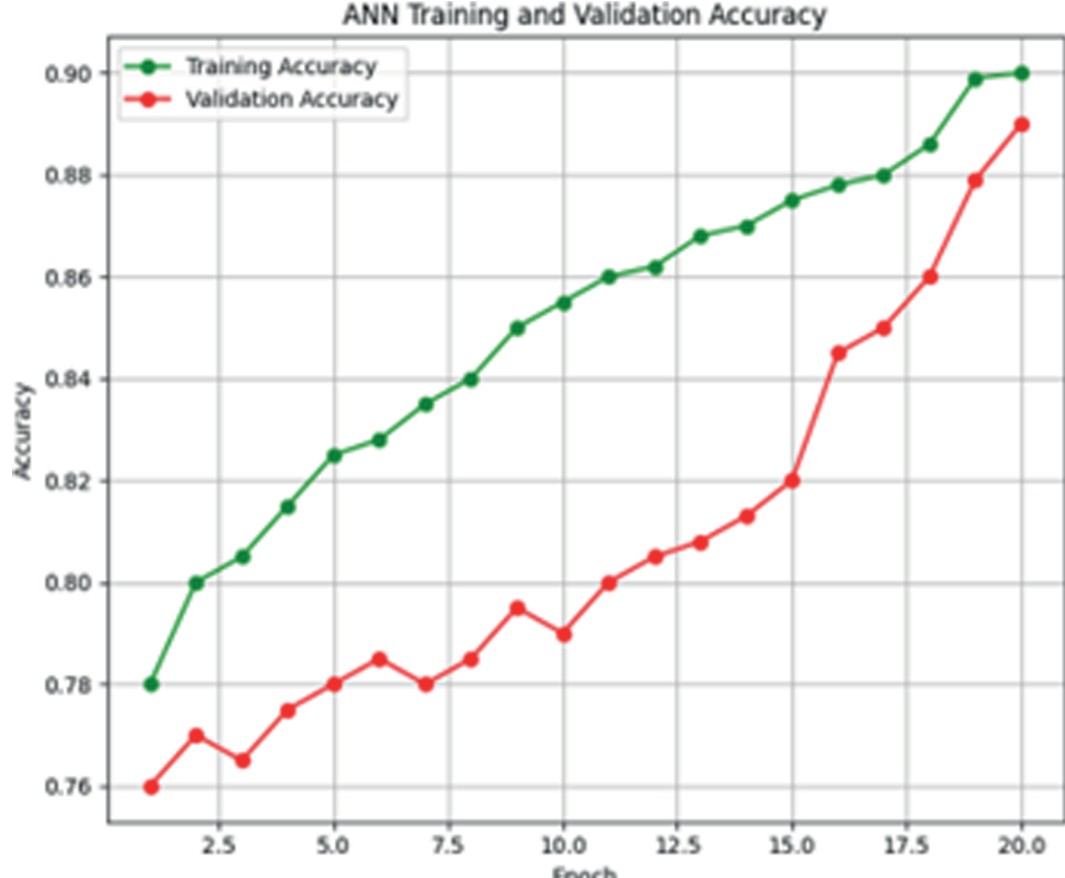

**Fig 9. Training VS. validation accuracy of ANN model.**

## 4.5 CNN model training

The next model for predicting heat events is CNN. This model is also trained on XWeather dataset with 1450 instances comprises a convolutional layer with 16 filters and ReLU activation function, followed by a max pooling layer for down-sampling. A flatten layer prepares data for dense layers, leading into a 10-neuron dense layer with ReLU activation. For binary classification, the final layer employs the Sigmoid function. Model training, using the Adam optimizer, was conducted over 20 epochs with a batch size of 32. Figs 10 and 11 shows the training and validation loss and accuracy graphs below.

## 4.6 LSTM model training

The third deep learning model in our proposed extreme heat prediction is LSTM. This model is trained on our XWeather dataset consisting of 1450 instances with a batch size of 32. The model utilized the softmax activation function and was optimized using the Adam optimizer. A stopping criterion was implemented, halting training if accuracy failed to improve for 2 consecutive epochs. Over the course of 50 epochs, the model concluded its training in the 20th epoch, achieving an accuracy of 96.2%. The graphs in Figs 12 and 13 are depicting the training and validation loss and accuracy.

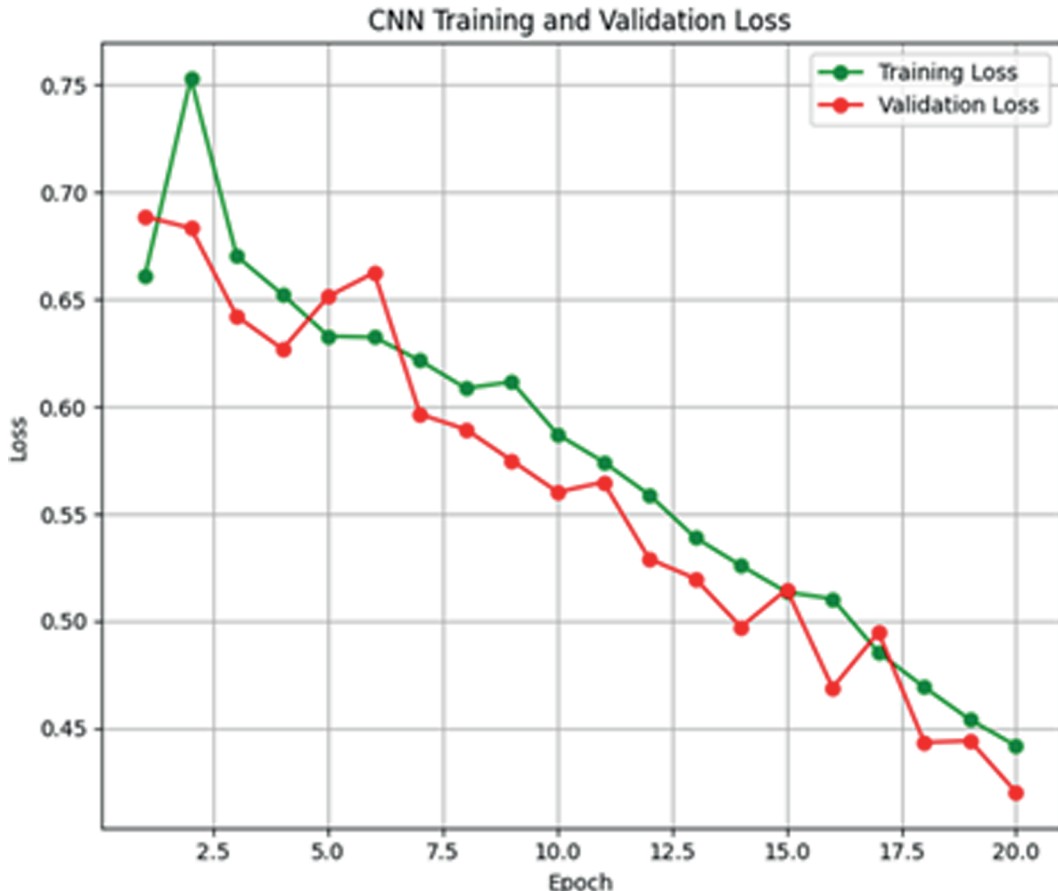

**Fig 10. Training vs. validation loss of CNN model.**

In our experiments, ANN, CNN, and LSTM models performed well for extreme heat prediction, but their black-box nature necessitated further investigation. To address this, we combined XAI methods, allowing us to interpret model predictions and identify critical factors influencing extreme heat predictions. This combination of deep learning and XAI improved accuracy and interpretability, allowing us to better understand model behavior and make more reliable predictions.

## 5 Results and discussions

In this study, the evaluation metrics for the deep learning (DL) models used to forecast weather data were accuracy, mean absolute error (MAE), mean squared error (MSE), and root mean squared error (RMSE). The differences between the three DL models are presented in the model comparison table (see Fig 14 and Table 4). With the lowest mean square error, lowest average mean square error, lowest root mean square error, and the highest accuracy among all models, the LSTM model was determined to be the most effective. The next stage involves using the LSTM model to predict extreme heat in the Lahore region based on a specific threshold. Subsequently, the most significant features in the XDL dataset were extracted using Explainable AI (XAI) techniques.

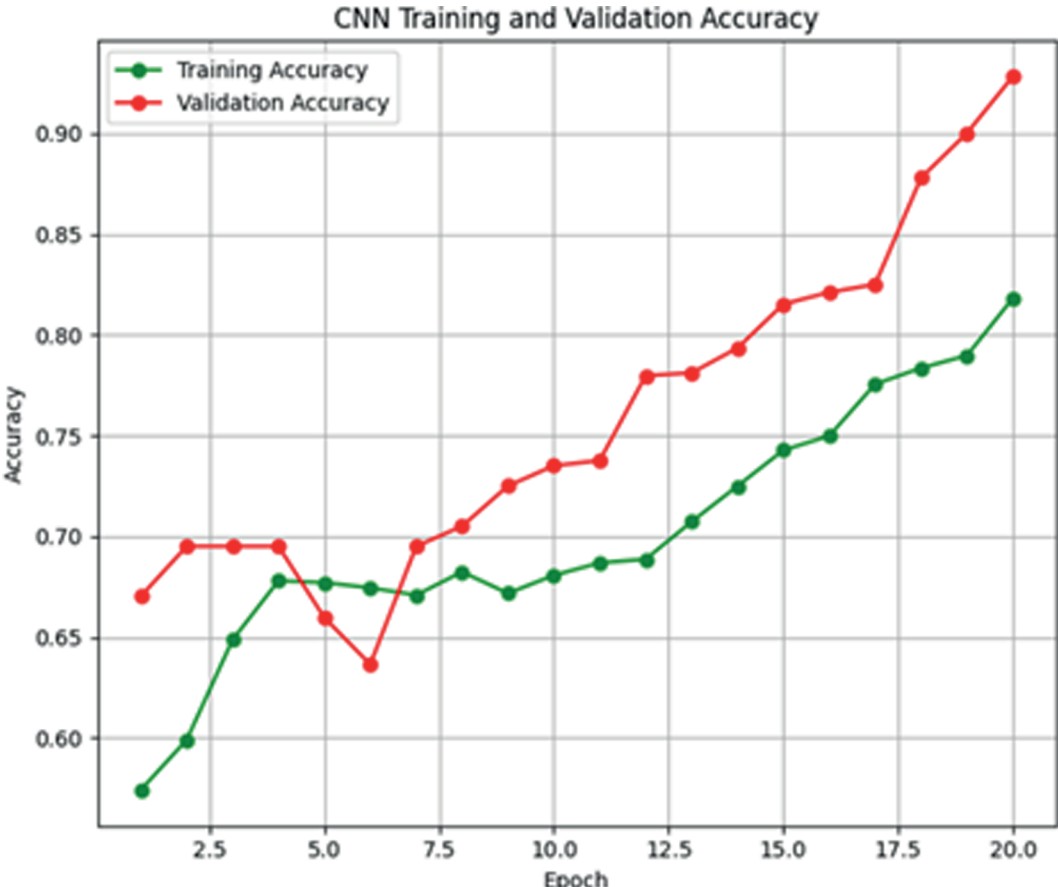

**Fig 11. Training vs. validation accuracy of CNN model.**

## 5.1 Regional threshold for extreme heat prediction

It is crucial to set the regional threshold percentile in order to provide accurate and context-specific weather predictions. For a given weather condition, this technique facilitate in risk assessment. Additionally, regional percentile for heat event prediction aids in the decision-making of meteorologists, decision-makers, and the general public. We determined the lead time of 1–3 days, utilizing a 95th percentile threshold for forecasting the maximum temperature. In weather data, 95th percentile means 95% of maximum temperature values fall above the threshold temperature. The calculated threshold temperature using XWeather dataset is 41 °C. Results (see Fig 15 and Table 5) after applying regional threshold to XWeather data are as under;

## 5.2 Interpretability of DL Model using SHAP and LIME models

After obtaining predictions from the LSTM model, we utilized XAI techniques to enhance the transparency of the black-box model. Fig 16 demonstrates the SHAP model's operation using XWeather data on LSTM model predictions. An overview of the results of the SHAP values for the anticipated outcome based on five years of data is shown in Fig 17. The prediction result was centered on the baseline, or SHAP value = 0.000. Correlated predictors would

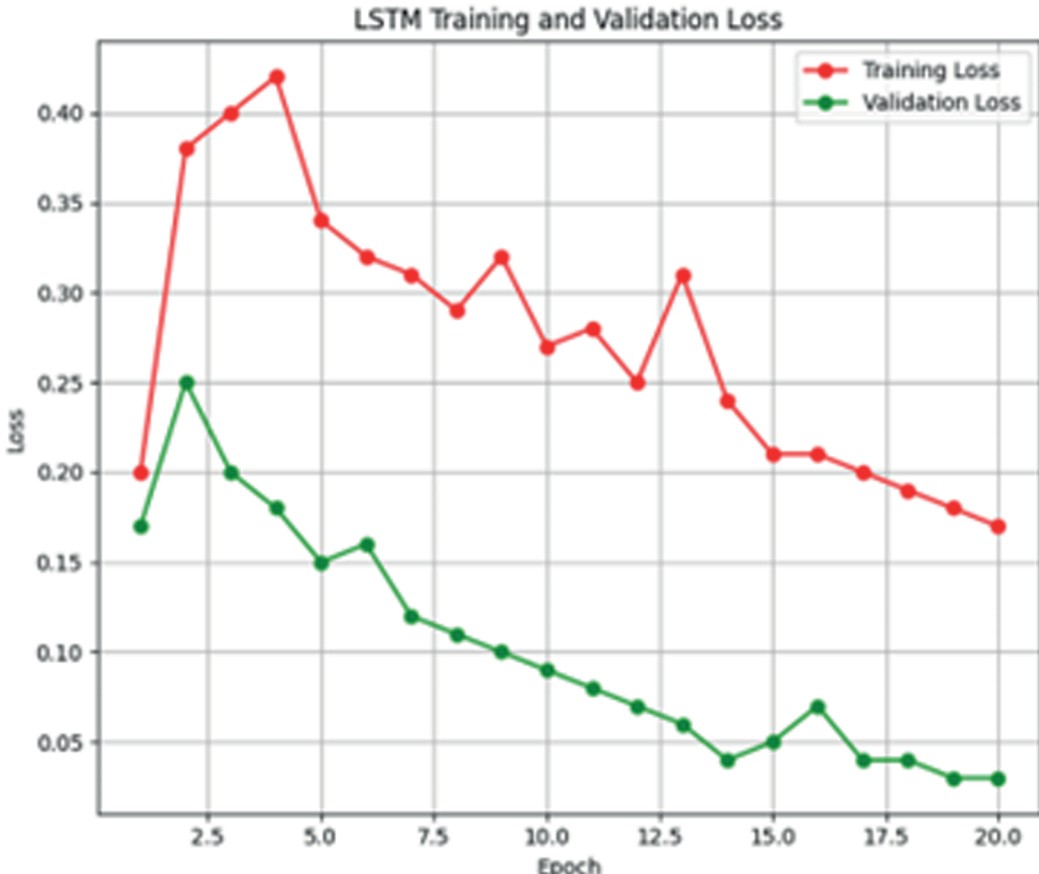

**Fig 12. Training vs. validation loss of LSTM model.**

not show how individual predictors affect prediction capability due to the additive nature of this characteristic [48].

Fig 17 illustrates the eleven parameters employed in locally interpretable framework i.e., LIME. Parameters crucial for making predictions, such as maximum temperature and humidity at 1200 UTC, are highlighted in orange, whereas less significant parameters are marked in blue.

The SHAP (see Fig 17) and LIME XAI (see Fig 18) models have revealed that humidity and maximum temperature stand out as the most influential parameters for predicting extreme heat events. By offering comprehensive and perceptive data, these two models' transparency and interpretability highlight the most essential features of forecasting.

## 5.3 Comparison with previous studies

The comparison of extreme heat prediction methods in Table 6 illustrates the performance and accuracy of various models and datasets. Study [49] uses the Integrated Surface Hourly (ISH) dataset on a Graph Neural Network (GNN) model, achieving 94.1 accuracy, 58.5 precision, and 62.5 recall. However, this study does not highlight the important factors of their dataset for predicting heatwaves. In Study [50], the Cheongju station (South Korea) dataset

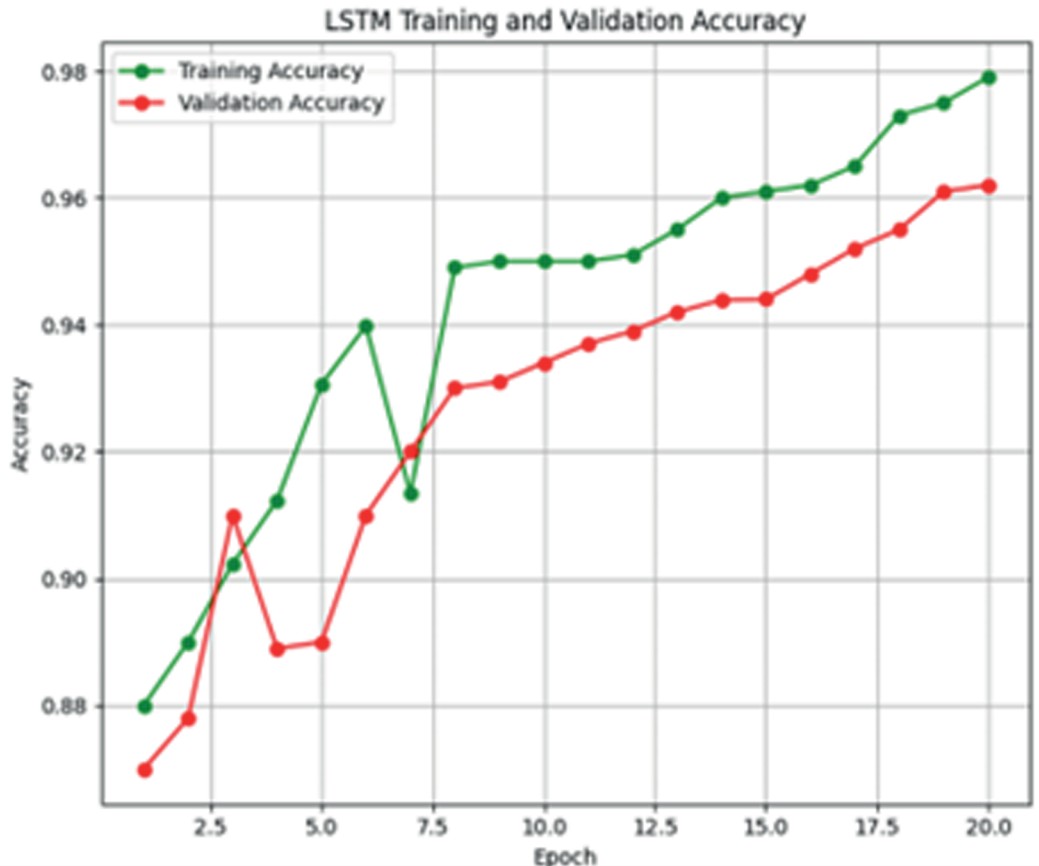

**Fig 13. Training vs. validation accuracy of LSTM model.**

of maximum temperature is used for predicting heatwaves using three deep learning models: ANN, LSTM, and RNN. Evaluation metrics include RMSE and MAE, but this study did not address lead time or the interpretability of the models. Study [51] employs the Weather and Climate dataset on Decision Tree (DT), CNN, and LSTM models, achieving accuracies of 80.5, 90.41, and 92.31 respectively, with the LSTM model scoring the highest in both accuracy and recall. In contrast, the proposed work uses the eXplainable Weather Dataset on deep learning models (ANN, CNN, and LSTM). The LSTM model scores the highest with short-term prediction and lead time, while also providing interpretability through XAI, identifying humidity and maximum temperature as the most important features for predicting extreme heat.

## 6 Conclusion

In this study, we proposed a deep learning-based framework to predict extreme heat events, using five years of XWeather data (2018–2022) for Lahore, collected from PMD. Among the deep learning models evaluated, the LSTM model demonstrated superior performance, achieving the lowest error and an accuracy of 96.2%. To enhance the interpretability of our model, we applied Explainable AI (XAI) techniques, specifically SHAP and LIME, which

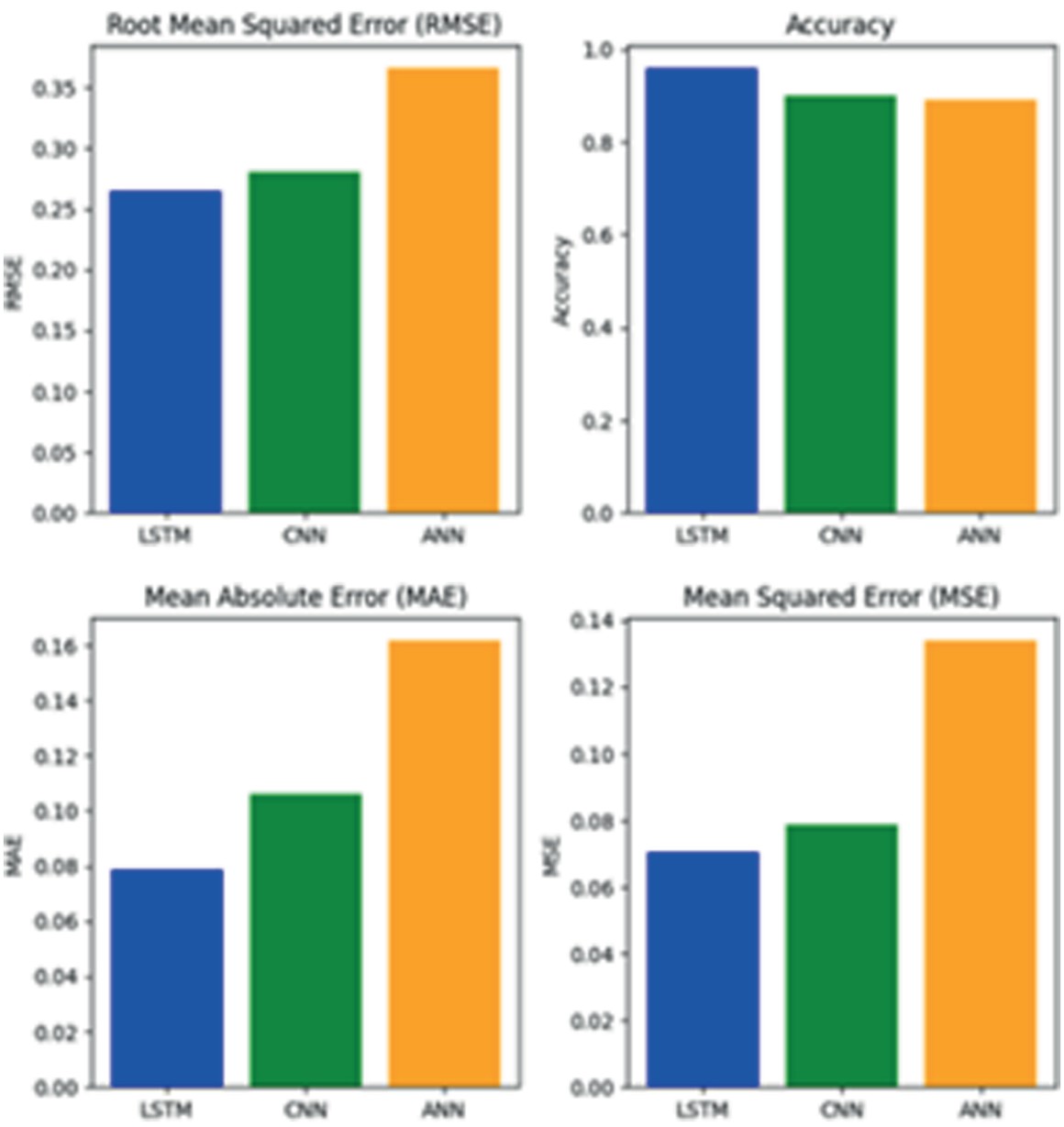

**Fig 14. Performance comparison of deep learning models.**

**Table 4. Comparison of models based on accuracy, MSE, MAE, and RMSE.**

| Model | Accuracy | MSE | | MAE | | RMSE | |
|---|---|---|---|---|---|---|---|
| | | Train | Test | Train | Test | Train | Test |
| ANN | 89.1 | 0.1379 | 0.1340 | 0.1824 | 0.1617 | 0.3691 | 0.3661 |
| LSTM | 96.2 | 0.8100 | 0.0704 | 0.7990 | 0.0787 | 0.2712 | 0.2653 |
| CNN | 92.5 | 0.0889 | 0.0787 | 0.1087 | 0.1058 | 0.2884 | 0.2804 |

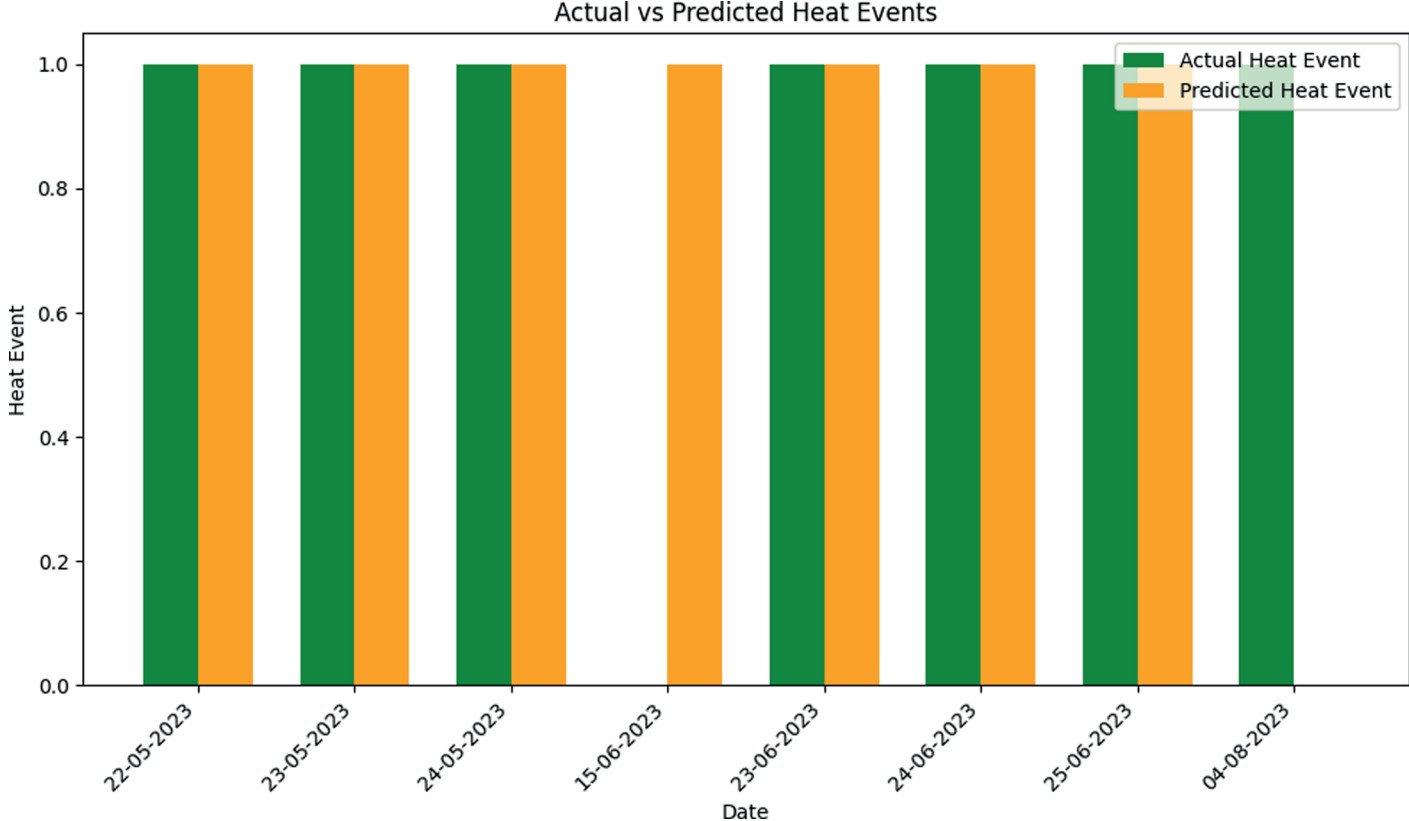

**Fig 15. Comparison of actual and predicted heat events: a deep learning perspective.**

**Table 5. Predicted extreme heat events and actual temperatures in Lahore.**

| Date | Heat Event Prediction | Extreme Heat Temperature (°C) |
|---|---|---|
| 22/05/2023 | 1 | 43.2 |
| 23/05/2023 | 1 | 42.6 |
| 24/05/2023 | 1 | 41.7 |
| 23/06/2023 | 1 | 41.2 |
| 24/06/2023 | 1 | 41.5 |
| 25/06/2023 | 1 | 41.6 |

provided valuable insights into the variables influencing our predictions. Our XAI analysis revealed that humidity and maximum temperature are the most critical factors driving accurate heatwave predictions, increasing the model's transparency and reliability.

Additionally, the use of regional percentiles, particularly the 95th percentile of maximum temperature, was found to be an effective tool for decision-making in extreme heat forecasting. For future improvements, we suggest incorporating data from desert regions like Al-Ahsa in Saudi Arabia to strengthen the model's performance. Further research should also focus on improving the scalability of deep learning and XAI techniques to handle larger datasets and diverse climate models. Incorporating additional data sources like satellite imagery and social media could further enhance prediction accuracy, allowing for a more adaptable and comprehensive model for various climatic conditions.

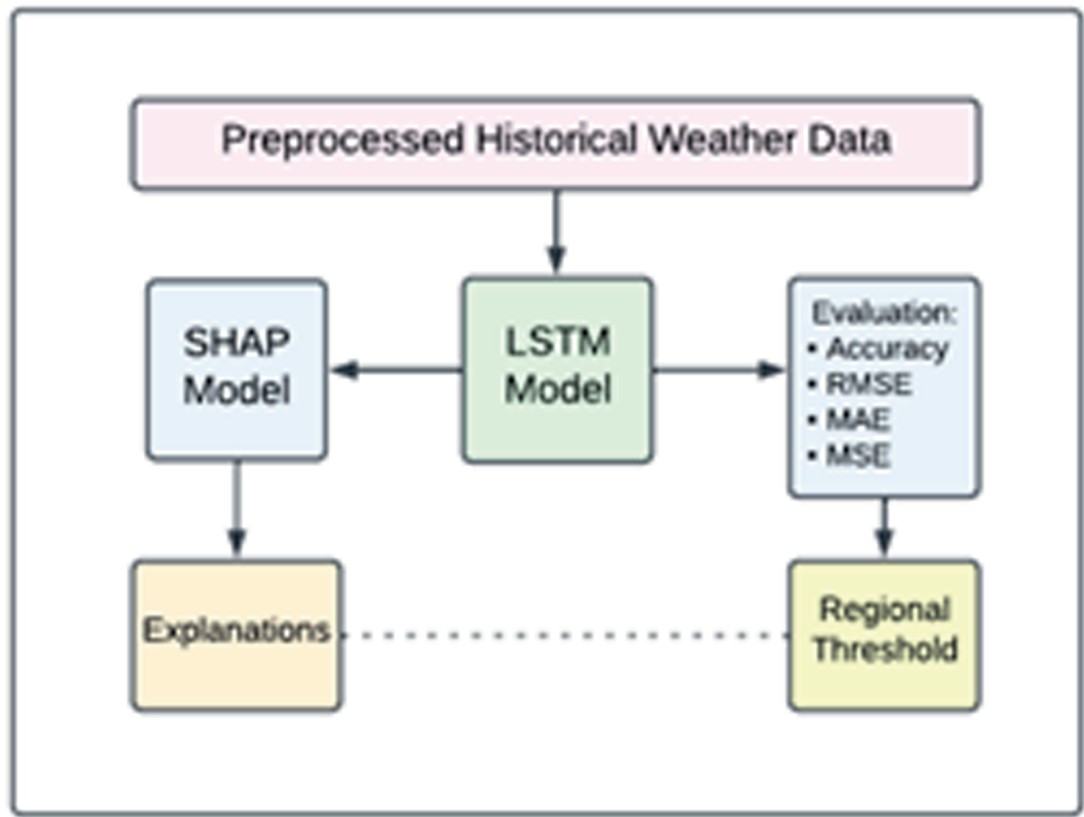

**Fig 16. Understanding the SHAP XAI process.**

**Table 6. Results of different models on various datasets.**

| Paper | Dataset | Model | Accu % | Preci % | Recall % | RMSE | MSE | MAE | XAI | Lead Time | Acc on our dataset |
|---|---|---|---|---|---|---|---|---|---|---|---|
| Peiyuan Li et al. [49] | Integrated Surface Hourly (ISH) | Graph Neural Network (GNN) | 94.1 | 58.5 | 62.5 | - | - | - | × | × | 69.89 |
| Trang Thi et al. [50] | Cheongju station (South Korea) | ANN | - | - | - | 2.425 | - | 1.917 | × | × | 96.2 with LSTM |
| | | RNN | - | - | - | 2.442 | - | 1.945 | | | |
| | | LSTM | - | - | - | 2.429 | - | 1.945 | | | |
| Sarita Bvagar et al. [51] | Weather and Climate | Decision Tree (DT) | 80.5 | 86.5 | 89.23 | - | - | - | × | 5 days | 96.2 with LSTM |
| | | CNN | 90.41 | 91.59 | 91.13 | - | - | - | | | |
| | | LSTM | 92.31 | 90.81 | 96.41 | - | - | - | | | |
| Proposed work | Explainable Weather Dataset | CNN | 92.5 | 90.21 | 89.86 | 0.2804 | 0.0787 | 0.1058 | ✓ | 1–3 days | 96.2 with LSTM |
| | | LSTM | 96.2 | 91.20 | 93.29 | 0.2653 | 0.0704 | 0.0787 | | | |

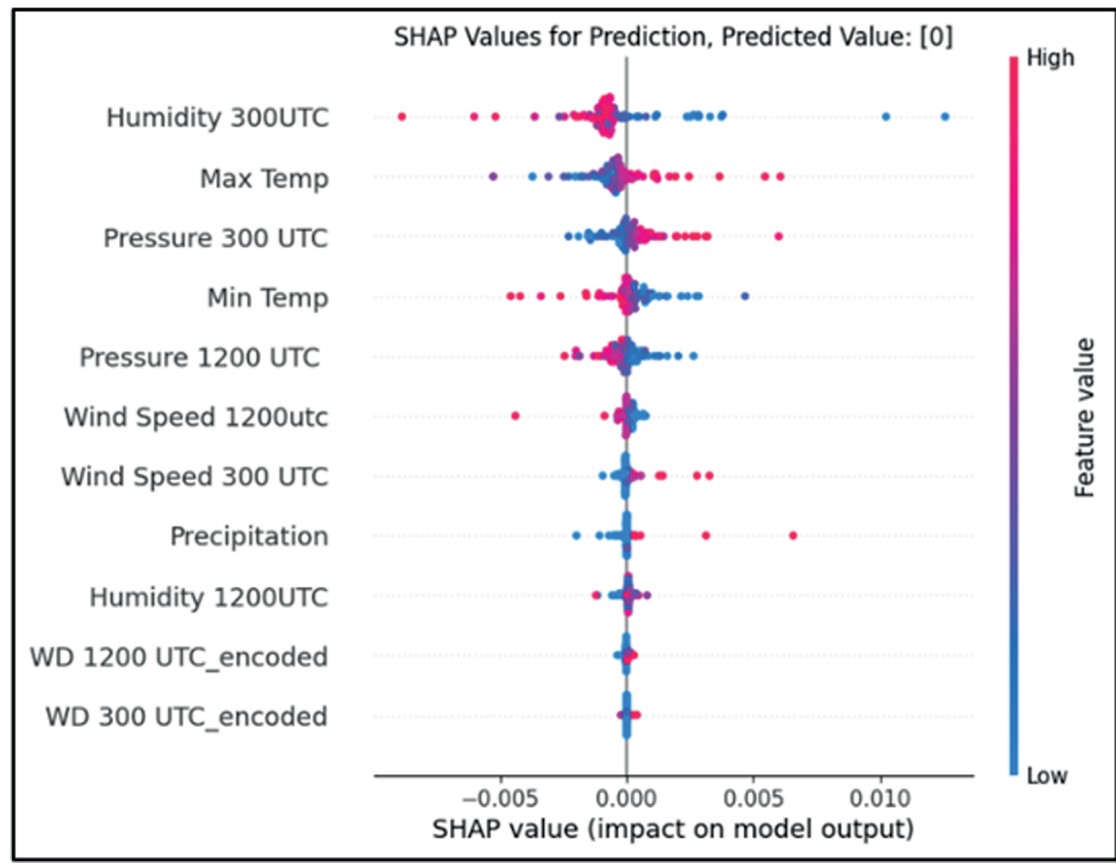

**Fig 17. Visualizing XAI SHAP model results for deep learning insight.**

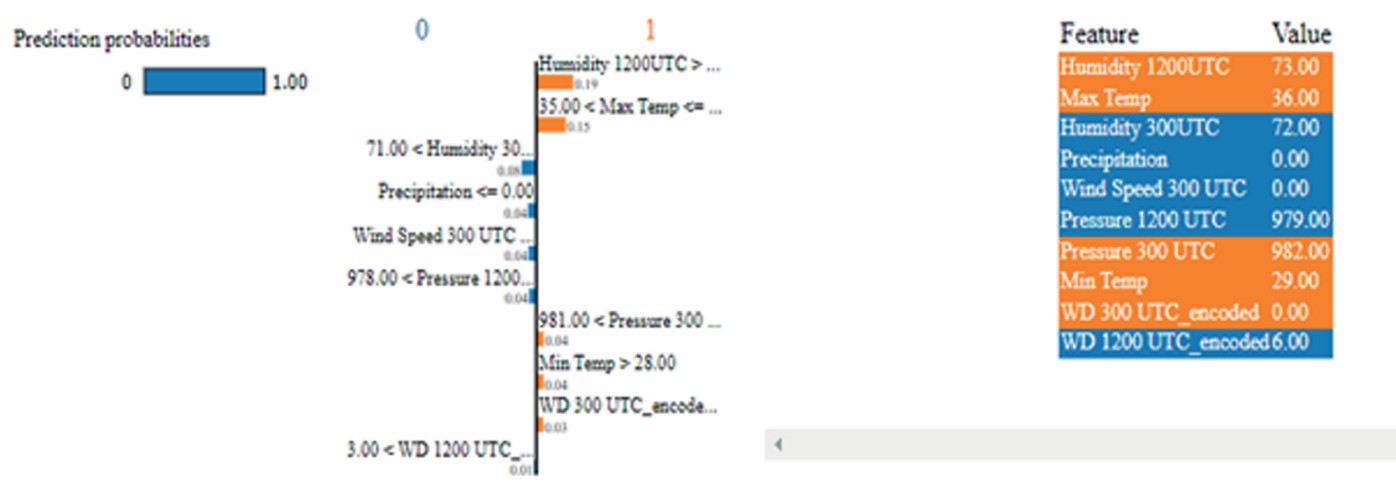

**Fig 18. Visual representation of XAI LIME model results.**

## Author contributions

**Conceptualization:** Amna Zafar.

**Data curation:** Fatima Shafiq, Sajid Iqbal.
**Formal analysis:** Fatima Shafiq.

**Funding acquisition:** Sajid Iqbal, Abdulmohsen Saud Albesher, Muhammad Nabeel Asghar.

**Investigation:** Fatima Shafiq, Amna Zafar.

**Methodology:** Fatima Shafiq, Amna Zafar.

**Project administration:** Muhammad Usman Ghani Khan.

**Resources:** Muhammad Usman Ghani Khan, Sajid Iqbal, Muhammad Nabeel Asghar.

**Supervision:** Muhammad Usman Ghani Khan.

**Writing – original draft:** Fatima Shafiq.

**Writing – review & editing:** Sajid Iqbal, Abdulmohsen Saud Albesher, Muhammad Nabeel Asghar

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
