## [Decision Letter · Decision Letter 0]

23 Oct 2024

PONE-D-24-39057Extreme heat prediction through deep learning and explainable AIPLOS ONE

Dear Dr. Iqbal,

Thank you for submitting your manuscript to PLOS ONE. After careful consideration, we feel that it has merit but does not fully meet PLOS ONE’s publication criteria as it currently stands. Therefore, we invite you to submit a revised version of the manuscript that addresses the points raised during the review process.

We look forward to receiving your revised manuscript.

Kind regards,

Upaka Rathnayake

Academic Editor

PLOS ONE

Journal Requirements:

"This work was supported by the Deanship of Scientific Research, Vice Presidency for 

Graduate Studies and Scientific Research, King Faisal University, Saudi Arabia, under 

Project GrantA468."

"NO authors have competing interests"

5. We note that you have indicated that there are restrictions to data sharing for this study. PLOS only allows data to be available upon request if there are legal or ethical restrictions on sharing data publicly. For more information on unacceptable data access restrictions, please see http://journals.plos.org/plosone/s/data-availability#loc-unacceptable-data-access-restrictions.

Reviewers' comments:

Reviewer's Responses to Questions

**Comments to the Author**

1. Is the manuscript technically sound, and do the data support the conclusions?

Reviewer #1: Yes

Reviewer #2: Yes

2. Has the statistical analysis been performed appropriately and rigorously? 

Reviewer #1: Yes

Reviewer #2: Yes

3. Have the authors made all data underlying the findings in their manuscript fully available?

Reviewer #1: No

Reviewer #2: Yes

4. Is the manuscript presented in an intelligible fashion and written in standard English?

Reviewer #1: Yes

Reviewer #2: Yes

5. Review Comments to the Author

Reviewer #1: The work presented can be credited and following are my comments

1.The results and discussion section does not have adequate depth and needs to be comprehensive and elaborated

2. Provide brief background on explainable AI methods and its usefulness for complex models. You may use

https://www.sciencedirect.com/science/article/pii/S235249282402275X

https://www.sciencedirect.com/science/article/pii/S2590123024001737

https://www.sciencedirect.com/science/article/pii/S2590123024007588

https://www.sciencedirect.com/science/article/pii/S266601642400313X

4. COnclusion needs to be rephrased by providing main findings of the study

5. What is the reasoning behind using LIME and SHAP both ?

6. In many of the works, LIME has been compared with SHAP and sometimes there are inconsistencies between LIME and SHAP

7. The description of SHAP and LIME should be in the methodology section and removed from results and dicussion

8. Results section should be significantly improved to have a scientifically thorough discussion.

9. Discussion should critically evaluate the predictions and explanations and compare findings and shortcomings with the related studies

Reviewer #2: Though this is good work, I am afraid about novelty this can be addressed by referring recent past papers/

i suggest few recent past papers.

Gene expression programming and artificial neural network to estimate atmospheric temperature in Tabuk, Saudi Arabia

HM Azamathulla, U Rathnayake, A Shatnawi

Applied Water Science 8, 1-7

Multivariate modeling of agricultural river water abstraction via novel integrated-wavelet methods in various climatic conditions

A Emadi, R Sobhani, H Ahmadi, A Boroomandnia, S Zamanzad-Ghavidel, ...

Environment, Development and Sustainability, 1-27

Recent climatic trends in Trinidad and Tobago, west indies

A Perera, SD Mudannayake, HM Azamathulla, U Rathnayake

Asia-Pacific Journal of Science and Technology 25 (2), 1-11

Assessment of climate change impact on the Gharesou River Basin using SWAT hydrological model

B Zahabiyoun, MR Goodarzi, ARM Bavani, HM Azamathulla

CLEAN–Soil, Air, Water 41 (6), 601-609

Impact of climate change on water resource engineering in trinidad and Tobago

A Chadee, M Narra, D Mehta, J Andrew, H Azamathulla

LARHYSS Journal P-ISSN 1112-3680/E-ISSN 2521-9782

Preserving fragile ecosystems from oil spills–An environmental sensitivity assessment of the east coast of Trinidad

C O'Brien-Delpesh, N Sinanan, H Martin, A Chadee

Ocean & Coastal Management 230, 106374

6. PLOS authors have the option to publish the peer review history of their article (what does this mean?). If published, this will include your full peer review and any attached files.

Reviewer #1: No

Reviewer #2: **Yes: **Mohammad Azamathulla Hazi

---

## [Author Response · Author response to Decision Letter 1]

7 Dec 2024

Please see attached "response to reviewers" file.

---

## [Decision Letter · Decision Letter 1]

10 Dec 2024

Extreme heat prediction through deep learning and

explainable AI

PONE-D-24-39057R1

Dear Dr. Iqbal,

We’re pleased to inform you that your manuscript has been judged scientifically suitable for publication and will be formally accepted for publication once it meets all outstanding technical requirements.

Kind regards,

Upaka Rathnayake

Academic Editor

PLOS ONE

Additional Editor Comments (optional):

Reviewers' comments:

Reviewer's Responses to Questions

**Comments to the Author**

1. If the authors have adequately addressed your comments raised in a previous round of review and you feel that this manuscript is now acceptable for publication, you may indicate that here to bypass the “Comments to the Author” section, enter your conflict of interest statement in the “Confidential to Editor” section, and submit your "Accept" recommendation.

Reviewer #1: All comments have been addressed

Reviewer #2: All comments have been addressed

2. Is the manuscript technically sound, and do the data support the conclusions?

Reviewer #1: Yes

Reviewer #2: Yes

3. Has the statistical analysis been performed appropriately and rigorously? 

Reviewer #1: Yes

Reviewer #2: Yes

4. Have the authors made all data underlying the findings in their manuscript fully available?

Reviewer #1: No

Reviewer #2: Yes

5. Is the manuscript presented in an intelligible fashion and written in standard English?

Reviewer #1: Yes

Reviewer #2: Yes

6. Review Comments to the Author

Reviewer #1: THe manuscript has been well revised. I accept

Reviewer #2: revision is appropriate

revision is appropriate

revision is appropriate

revision is appropriate

revision is appropriate

revision is appropriate

revision is appropriate

revision is appropriate

7. PLOS authors have the option to publish the peer review history of their article (what does this mean?). If published, this will include your full peer review and any attached files.

Reviewer #1: No

Reviewer #2: **Yes: **Hazi Azamathulla

---

## [Editor Report · Acceptance letter]

PONE-D-24-39057R1

PLOS ONE

Dear Dr. Iqbal,

I'm pleased to inform you that your manuscript has been deemed suitable for publication in PLOS ONE. Congratulations! Your manuscript is now being handed over to our production team.

Kind regards,

on behalf of

Professor Upaka Rathnayake

Academic Editor

PLOS ONE